# Glucose Metabolism in Pancreatic Cancer

**DOI:** 10.3390/cancers11101460

**Published:** 2019-09-29

**Authors:** Liang Yan, Priyank Raj, Wantong Yao, Haoqiang Ying

**Affiliations:** 1Department of Molecular and Cellular Oncology, University of Texas MD Anderson Cancer Center, Houston, TX 77030, USA; yliang4@mdanderson.org (L.Y.); praj@mdanderson.org (P.R.); 2Department of Translational Molecular Pathology, University of Texas MD Anderson Cancer Center, Houston, TX 77030, USA; wyao2@mdanderson.org

**Keywords:** glucose metabolism, pancreatic cancer

## Abstract

Pancreatic ductal adenocarcinoma (PDAC) is one of the most aggressive and lethal cancers, with a five-year survival rate of around 5% to 8%. To date, very few available drugs have been successfully used to treat PDAC due to the poor understanding of the tumor-specific features. One of the hallmarks of pancreatic cancer cells is the deregulated cellular energetics characterized by the “Warburg effect”. It has been known for decades that cancer cells have a dramatically increased glycolytic flux even in the presence of oxygen and normal mitochondrial function. Glycolytic flux is the central carbon metabolism process in all cells, which not only produces adenosine triphosphate (ATP) but also provides biomass for anabolic processes that support cell proliferation. Expression levels of glucose transporters and rate-limiting enzymes regulate the rate of glycolytic flux. Intermediates that branch out from glycolysis are responsible for redox homeostasis, glycosylation, and biosynthesis. Beyond enhanced glycolytic flux, pancreatic cancer cells activate nutrient salvage pathways, which includes autophagy and micropinocytosis, from which the generated sugars, amino acids, and fatty acids are used to buffer the stresses induced by nutrient deprivation. Further, PDAC is characterized by extensive metabolic crosstalk between tumor cells and cells in the tumor microenvironment (TME). In this review, we will give an overview on recent progresses made in understanding glucose metabolism-related deregulations in PDAC.

## 1. Introduction

Pancreatic ductal adenocarcinoma (PDAC) is one of the most aggressive solid malignancies, which is projected to soon become the second leading cause of cancer-related deaths in the US [1]. Oncogenic Kirsten rat sarcoma 2 viral oncogene homolog (KRAS) mutations, which occur in over 90% of human PDAC, are the dominant driver for tumor progression and play a critical role in reprogramming metabolism [2]. It has been known for almost a century that cancer cells take up enormous amounts of glucose, which is fermented to produce lactate even in the presence of oxygen, a process described as the Warburg effect. The development of various sophisticated analytic tools, including mass spectrometry- or nuclear magnetic resonance-based (NMR-based) metabolomic analysis, isotype-labeled nutrient tracing, and hyperpolarized magnetic resonance metabolic imaging in patients and preclinical models, has greatly enhanced the comprehensive and in-depth characterization of tumor metabolism programs. As a result, the reprogramming of metabolic pathways, including enhanced glycolysis, has been recognized as one of the emerging hallmarks of cancer [3]. Glycolysis is the central carbon metabolism pathway in cells, which provides energy in the form of ATP and also fuels cell growth and division by providing biomass. Importantly, there are several critical metabolic pathways that branch from glycolysis, including the pentose phosphate pathway (PPP), hexosamine biosynthesis pathway (HBP), serine biosynthesis, and tricarboxylic acid cycle (TCA cycle). These branched pathways have been proven to work together or individually to promote tumorigenesis.

The critical role of reprogramed metabolism has been well characterized in PDAC tumorigenesis. KRAS mutation in PDAC can enhance the glycolytic pathway by upregulating the expression of glucose transporters and rate-limiting enzymes of glycolysis, such as hexokinase 2 (HK2), phosphofructokinase-1 (PFK1), and lactate dehydrogenase A (LDHA) [2,4]. The enhanced glucose metabolism can promote PDAC tumorigenesis by providing energy (glycolytic flux and TCA cycle), new biomass support (PPP and serine biosynthesis pathway), reactive oxygen species (ROS) maintenance (glutamine metabolism and TCA cycle), signal modulation (HBP), and DNA methylation (serine biosynthesis pathway). In addition, oncogenic KRAS also reprograms the glutamine metabolism to support the redox homeostasis of PDAC [5]. Furthermore, a broad metabolite profiling analysis characterized three different metabolic subtypes of PDAC, including the slow proliferating, glycolytic, and lipogenic subtype. Lipogenic PDAC tumors are associated with the epithelial phenotype, whereas the glycolytic subtype was related to the mesenchymal phenotype. Notably, different subtypes of PDAC classified on the basis of their metabolic profile show distinct sensitivities to metabolic inhibitors, which illustrates the metabolic heterogeneity of PDAC [6]. In addition, cancer cells can enhance salvage pathways, such as autophagy [7] and macropinocytosis [8,9], to further scavenge essential metabolites to support its fast division or proliferation. Accumulating evidence has established that the metabolic reprogramming in the tumor environment is also critical for PDAC tumorigenesis. Metabolites released from stromal cells can be taken up by cancer cells to support tumor growth or may be used to generate resistance to drugs [10,11]. The translational potential of targeting metabolism pathways for cancer treatment has been nicely reviewed recently [12]. Here, we will provide a broad description of the molecular mechanisms underlying the glucose metabolism reprogramming in PDAC.

## 2. Activation of Glycolysis in PDAC

Altered glycolysis has been recognized as the major metabolic alteration in pancreatic cancer. The glycolytic flux in cancers is tightly controlled to meet the fast proliferative needs, as well as to provide building blocks for synthetic reactions. As such, many glycolytic enzymes are associated with poor prognosis of PDAC. Genetic or pharmacological inhibition of key glycolytic genes, such as HK2, PFK1, and LDHA, has been shown to inhibit the tumorigenic activity of PDAC cells. Here, we will focus on the mechanisms that maintain a high glycolysis rate in PDAC (Figure 1).

### 2.1. Enhanced Glucose Uptake

The transport of glucose across the plasma membrane is one of the rate-limiting steps and is mediated by two classes of glucose transporters, including facilitated transporters (GLUTs) and active transporters or symporters (SGLTs). Facilitative GLUTs enable the ATP-independent transportation of glucose across a hydrophobic cell membrane, down its concentration gradient. Among the GLUTs, expression of GLUT-1 (SLC2A1) is associated with PDAC progression. A progressive increase in the expression of GLUT-1 was observed from low-grade to higher-grade dysplastic lesions, with no expression in the acini or ducts in normal pancreas and detectable expression in 74% of cases [13], which may be due to the activation of mutant KRAS [2]. The increased expression of GLUT-1 in human pancreatic tumors was also suggested by the higher rate of 18F-FDG uptake into tumor cells compared with normal pancreatic tissue [14,15]. In addition, a direct correlation was observed between GLUT-1 expression and histological grade or tumor size in PDAC patients [13]. Patients with low expression of GLUT-1 in the primary tumor have a better prognosis and therapeutic response to neoadjuvant chemoradiotherapy compared with those with high GLUT-1 expression [16], underscoring the role of GLUT-1 in PDAC malignancy. Besides the transcriptional regulation, it has been shown that Paraoxonase 2 (PON2) directly interacts with GLUT-1 to stimulate glucose uptake and therefore promote PDAC tumor growth [17]. Interestingly, PON2 increased glucose uptake via mediation of the interaction of stomatin and GLUT1 rather than transcriptional or translational regulation of GLUTs or HK2.

In additional to GLUTs, a recent study has indicated that PDAC also overexpresses an alternative glucose transporter, SGLT2 (SLC5A2), which is normally expressed in the kidney, and mediates sodium-dependent glucose reabsorption [18,19]. Treating PDAC xenograft models with canagliflozin, a SGLT2 inhibitor used for type 2 diabetes, not only inhibited tumor growth but also sensitized tumors to PI3K inhibitor [19,20]. However, it remains to be further investigated whether the anti-tumor response of canagliflozin is due to the direct impact on tumor cell SGLT2 or systematic effect on the blockade of feed-back insulin induction.

### 2.2. Feedback Regulation of Glycolysis

There are a total of 10 steps in glycolytic process linking extracellular glucose to excreted lactate, of which three steps catalyzed by hexokinase, phosphofructokinase, and pyruvate kinase are virtually irreversible and are believed to be the regulatory sites in the glycolytic flux. Among them, phosphofructokinase (PFK), which catalyzes the rate-limiting phosphorylation of fructose-6-phosphate to fructose-1,6-bisphosphate, serves as the gatekeeper for mammalian glycolysis and is subjected to allosteric inhibition by ATP. The inhibitory effect of ATP is further potentiated by a decrease in pH [21]. To maintain glycolysis at a high rate, cancer cells need to maintain the homeostasis of intra-cellular pH levels by actively transporting lactate into the extracellular space. Indeed, monocarboxylate transporter 1 (MCT1) and monocarboxylate transporter 4 (MCT4), two major lactate transporters, are robustly expressed in PDAC cells [22]. KRAS signaling has been shown to induce the expression of MCT4 to promote lactate efflux and thus mitigate the toxic effects of intracellular lactate accumulation due to elevated glycolysis [23]. Besides the pH regulation, export of lactate will promote the reduction of pyruvate to lactate and oxidation of nicotinamide adenine dinucleotide (NADH) to NAD+. Importantly, the generated NAD+ is an essential coenzyme for the oxidation of glyceraldehyde 3-phosphate and maintenance of continued glycolysis. In addition, the membrane localization of MCT1 and MCT4 has been shown to be facilitated by cluster of differentiation 147 (CD147), a glycoprotein highly expressed in PDAC [24,25]. CD147 depletion results in the accumulation of lactate in tumor cells and suppressed PDAC growth in xenograft models [26]. Moreover, a recent study indicated that lactate may also be dissipated through gap junctions besides the exportation through lactate transporters [27].

Another factor that augments the inhibitory effect of ATP on phosphofructokinase is citrate, an intermediate of the TCA cycle [28]. Citrate is exported out of mitochondria and utilized by ATP-citrate lyase (ACLY) for the maintenance of the cytosolic acetyl coenzyme A (acetyl-CoA) pool. Therefore, enhanced ACLY activity in tumor cells serves to avoid the accumulation of citrate to ensure glycolysis flux and cell proliferation. Indeed, ACLY-dependent metabolism is elevated during PDAC development, and deletion of ACLY suppressed tumor growth [29]. Interestingly, while ACLY depletion leads to inhibition of glycolysis as expected, citrate is not elevated, at least in ACLY knockout adipocytes, implicating additional mechanisms for the regulation of glycolysis [30].

Besides the regulation through allosteric inhibition, the activity of PFK is also tightly controlled by its allosteric activator, fructose 2,6-bisphosphate, whose metabolism is mediated by the family of bifunctional enzymes 6-phosphofructo-2-kinase/fructose 2,6-bisphosphatases (PFKFBs). Among them, PFKFB3 is regarded as the major player contributing to elevated glycolysis in tumor cells due to its unique high kinase/phosphatase activity ratio and the inducible nature of this gene under hypoxia and inflammatory conditions [31,32,33]. Indeed, PFKFB3 is overexpressed in a variety of human cancers, including pancreatic cancer [34,35]. Moreover, PFKFB3 is required for RAS-induced cellular transformation [36], underscoring the importance of PFKFB3-mediated glycolysis regulation during tumor development.

Such feedback regulation of glycolysis has also been exploited for the inhibition of additional regulatory steps in glycolysis to target PDAC cells. A recent study showed that high mannose treatment leads to inhibition of tumor growth in mannose phosphate isomerase (MPI)-low pancreatic cancer cells [37]. MPI catalyzes the conversion between fructose-6-phosphate and mannose-6-phosphate and high dose mannose treatment in MPI-low cells leads to the accumulation of mannose-6-phosphate, which in turn shuts down glycolysis and suppresses tumor growth through the feedback inhibition of glycolytic enzymes hexokinase (HK) and phosphoglucose isomerase (PGI) [37,38].

### 2.3. Transcriptional and Post-Transcriptional Control of Glycolysis

The glycolytic activity in tumor cells is regulated in part by the expression of rate-limiting glycolytic genes. The major determinants for the expression level of glycolysis genes in PDAC include its driver oncogene and the tumor microenvironment. It is well known that oncogenic KRAS induces glycolysis and recent studies demonstrated that KRAS signaling plays a profound role in the transcription of glucose transporters and key glycolysis genes in tumor cells [2,39,40]. As a key dependence in KRAS-driven tumors, the MYC oncogene has been implicated in the induction of glycolysis genes downstream of KRAS signaling in PDAC cells [2,41,42,43]. One defining feature of human PDAC is its severely hypoxic microenvironment [44]. Under hypoxia, hypoxia-inducible factor (HIF1A) is stabilized and induces the expression of multiple genes involved in glycolysis [45,46]. Accordingly, HIF1A levels are upregulated in PDAC, and multiple glycolysis-related genes, including GLUT-1 and MCT4, are highly expressed in the hypoxic regions [47]. It should be noted that oncogenic KRAS can induce HIF1A in a PI3K-dependent manner independent of hypoxia [48,49]. How KRAS signaling may cooperate with hypoxia to induce glycolysis and the relationship between MYC-driven and HIF1A-driven expression of glycolysis genes in PDAC remain to be fully elucidated. Both MUC1 and MUC13 were reported to stabilize the HIF1 in PDAC, which subsequently enhances glucose metabolism by elevated expression of glycolysis enzymes [50,51,52]. While additional transcription factors, such as KLF4 and FOXM1, have also been reported to induce glycolysis in PDAC cells through the transcriptional upregulation of LDHA [53,54], their in vivo relevance needs to be further validated.

In addition, the expression of glycolysis genes is also regulated at the post-transcriptional level. BCL2 associated athanogene 3 (BAG3), a cochaperone protein, has been shown to stabilize HK2 mRNA and therefore induce HK2 expression to enhance glycolysis and tumorigenic activity in PDAC cells [55]. Accordingly, BAG3 is found to be overexpressed in PDAC and correlated with poor clinical outcome in patients [55,56]. HK2 expression is also negatively regulated by microRNA-143 (mir-143), a tumor suppressive microRNA (miRNA) [57,58]. Importantly, miR-143 expression has been shown to be silenced by oncogenic KRAS in PDAC cells [59]. In addition, miR-34a is another tumor suppressive miRNA likely involved in the glycolytic regulation in PDAC. MiR-34a, a downstream target of tumor suppressor p53, is downregulated in the majority of PDAC cells and its low expression is significantly correlated with the poor prognosis of PDAC patients [60,61]. Similar to miR-143, miR-34a has been implicated in the regulation of glycolysis genes, including HK1, HK2, GPI, and LDHA [62,63]. Therefore, driver mutations of PDAC, such as KRAS and TP53, are also involved in the post-transcriptional regulation of glycolysis.

Besides regulation at the expression level, multiple glycolysis genes are also modulated at the post-translational level in PDAC. Hypoxia has been shown to activate PFKFB3 through AMPK-dependent phosphorylation [62,64]. This is concordant with the critical role of AMPK for the adaptation of PDAC cells under conditions of metabolic stress [65]. Additionally, the deacetylase SIRT2 has been shown to be overexpressed in PDAC, which results in LDHA deacetylation and activation to promote glycolysis in tumor cells [66]. Moreover, tumor suppressor FBW7 has been shown to function as a negative regulator of glucose metabolism via the c-Myc/TXNIP axis in PDAC [67]. Interestingly, hyperactivation of RAS–RAF–MEK–ERK signaling decreases FBW7 expression in PDAC [68], indicating the role of FBW7 in KRAS-regulated glucose metabolism.

Lastly, a recent study demonstrated that PI3K-mediaiated Rac1 activation leads to the disruption of actin filaments, which mobilizes aldolase from the cytoskeleton and elevates the enzymatic activity of aldolase to increase glucose flux through glycolysis [69]. Notably, the PI3K-Rac1 axis plays a critical role for the cytoskeleton remodeling in PDAC and is required for KRAS-induced pancreatic tumorigenesis [70,71]. These studies implicate the important role of subcellular localization in the regulation of glycolytic enzyme activities in response to oncogenic signaling in PDAC.

In summary, the regulation of glycolysis in PDAC is largely achieved through epigenetic and post-transcriptional mechanisms. However, it does not exclude the possibility that alterations at the genetic level are also contributing to glycolysis activation in PDAC even though recurrent somatic alterations in glycolysis genes are rare in PDAC if not none. Indeed, single nucleotide polymorphisms in HK2 have been reported to be significantly associated with patient survival in PDAC [72], prompting the need to further validate the function of those gene variants and their involvement in metabolic regulation in PDAC.

## 3. Enhanced Glucose Flux into Anabolic Pathways

### 3.1. Pentosephosphate Pathway (PPP)

The PPP is composed of two arms: The oxidative arm and the non-oxidative arm. Both arms utilize glycolysis intermediates for the synthesis of ribose-5-phosphates while only the oxidative one generates NAPDH. The PPP is crucial to support the rapid and uncontrolled growth of cancer cells by producing both ribonucleotides for DNA/RNA building and the essential cofactor NADPH for ROS detoxification and macromolecule biosynthesis.

The oxidative PPP has been shown to be activated by oncogenic KRAS during cellular transformation and function to promote cellular proliferation, likely due to its ROS scavenger function through the generation of NADPH [73,74]. However, a recent study in a KRAS-driven mouse PDAC model indicates that the flux through oxidative PPP does not seem to be dependent on KRAS signaling [2]. It is possible that oxidative PPP in PDAC is activated due to p53 mutation since p53 has been reported to suppress the activity of oxidative PPP through direct interaction with glucose-6-phosphate dehydrogenase (G6PD), the key enzyme for oxidative PPP [75]. Contradictorily, p53 has also been shown to enhance oxidative PPP activity through the induction of TP53-inducible glycolysis and apoptosis regulator (TIGAR), which in turn inhibits glycolysis through its fructose bisphosphatase activity and promotes the flux of accumulated G6P through oxidative PPP [76]. Consistent with this notion, p53 mutant PDAC cells with low TIGAR expression seem particularly dependent on glycolysis and exhibit enhanced apoptosis following glycolysis inhibition with LDHA inhibitor [77]. Moreover, a recent study demonstrated that GOT1-mediated glutamine metabolism, instead of the oxidative PPP, plays a major role for NADPH generation and ROS homeostasis in PDAC cells [5]. While the requirement of the oxidative PPP-NADPH axis in advanced PDAC remains to be further evaluated, recent studies indicate that 6-phosphogluconate dehydrogenase (PGD), an enzyme of the oxidative arm of PPP, is specifically activated in metastatic PDAC [78]. PGD depletion selectively inhibits the tumorigenic activity of metastatic tumor clones [78,79]. Interestingly, the function of PGD in metastatic PDAC seems outside of the canonical oxidative PPP and involves a metabolism route similar to the pentose conversion pathway that converts glucose to PGD substrate 6-phosphogluconate (6PG) through a series of enzymes, including UDP-Glucose-6-Dehydrogenase (UGDH), Aldo-Keto Reductase 1A1 (AKR1A1), and gluconokinase (IDNK) [79] (Figure 2).

In contrast to the uncertainty surrounding the role of oxidative PPP, accumulating evidence indicates that the non-oxidative arm is highly active in PDAC to promote tumor growth [80,81]. Moreover, it has been shown that oncogenic KRAS selectively activates non-oxidative PPP, likely through the induction of genes in the non-oxidative arm, such as ribulose-5-phosphate isomerase (RPIA) and ribulose-5-phosphate-3-epimerase (RPE) [2,82]. Depletion of RPIA significantly inhibited the tumorigenic activity of KRAS-driven PDAC cells, underscoring the importance of non-oxidative PPP during tumor maintenance [2,82]. Notably, elevated pyrimidine biosynthesis due to enhanced glucose flux through non-oxidative PPP has been shown to contribute to gemcitabine resistance in PDAC, and reduced expression of transketolase (TKT), a non-oxidative PPP enzyme, is correlated with increased gemcitabine sensitivity in PDAC patients [52], further supporting the translational potential of targeting non-oxidative PPP.

### 3.2. Hexosamine Biosynthesis Pathway (HBP)

HBP uses substrates, such as fructose-6-phosphate, acetyl-CoA, glutamine, and UTP, to generate UDP-GlcNAc (N-acetylglucosamine), therefore integrating multiple metabolism pathways, including glucose, amino acid, and lipid metabolism. The generated UDP-GlcNAc is used for glycosylation of proteins and lipids, or conversion to UDP-GlcNAc-derived activated monosaccharides, such as UDP-GalNAc and CMP-Neu5Ac, which are also used for glycosylation [83].

A recent study in a PDAC genetically engineered mouse (GEM) model indicated that oncogenic KRAS signaling significantly increases the flux of glycolysis intermediates through HBP by inducing the expression of glucosamine-fructose-6-phosphate aminotransferase 1 (GFPT1), the rate-liming enzyme for HBP [2]. Such induction of GFPT1 expression and enhanced glucose flux through HBP is further enhanced under hypoxia conditions to maintain cell viability [84]. GFPT1 knock down in PDAC cells strongly suppresses tumorigenic activity in vitro and in vivo, indicating the crucial role of HBP during the maintenance of PDAC [2]. Correspondingly, high expression of GFPT1 has been shown to be significantly correlated with poor prognosis in PDAC patients [85].

UDP–GlcNAc, the end product of HBP, is the limiting substrate for the O-linked GlcNAc modifications (O-GlcNAcylation) of proteins. Consistent with the enhanced HBP activity, protein O-GlcNAcylation is induced by KRAS oncogene as well as hypoxia in PDAC cells [2,84]. Hyper-O-GlcNAcylation in human PDAC is also associated with elevated expression of O-linked GlcNAc transferase (OGT), a key enzyme for protein O-GlcNAcylation, and reduction of O-GlcNAcase (OGA), an enzyme that removes O-GlcNAc [86]. O-GlcNAcylation has been recognized as a major type of post-translational modification, which functions to sense nutrient availability and regulate key oncoproteins, such as Nuclear Factor Kappa B (NF-κB) and Yes-associated protein 1 (YAP1), to support tumor growth [86,87]. In addition, HBP-mediated O-GlcNAcylation may also function to finetune the glucose flux in tumor cells. It was recently reported that a key glycolysis enzyme, PFK-1, is also modified with O-GlcNAcylation, which suppresses PFK-1 activity and redirects the flow of glucose metabolism toward PPP during hypoxia [88], providing a mechanism for the coordination between anabolic branches of glucose metabolism in tumor cells (Figure 2).

### 3.3. Serine Biosynthesis Pathway

As a central node for the biosynthesis of many molecules, serine biosynthesis is critical in various cancer types [89,90,91,92]. While some tumor types rely on exogenous serine, PDAC tissues are particularly low in serine [93]. On the other hand, serine biosynthesis genes are induced by oncogenic KRAS, which re-wire the PDAC metabolism to depend on de novo serine biosynthesis for survival [94]. Serine is generated de novo from 3-phosphoglycerate, a glycolysis intermediate, and is the precursor of the nonessential amino acids glycine and cysteine. Both glycine and cysteine are precursors of glutathione, which are critical for the maintenance of ROS levels in the cell. Serine metabolism is also critical for the mitochondrial redox homeostasis during hypoxia. Upon hypoxia stress, the mitochondrial enzyme SHMT2 is induced by HIF1A, and is critical for NADPH production through serine catabolism, which in turn maintains redox balance to support tumor cell survival and growth [91]. In addition, serine supplies carbon to the one-carbon pool and is involved in both DNA and histone methylation (Figure 2). Oncogenic KRAS has been shown to corporate with LKB1 loss to induce the serine–glycine–one-carbon pathway to fuel tumor growth. Accordingly, human PDAC cells with LKB1 mutations are more sensitive to the inhibition of serine biosynthesis [92].

### 3.4. Mitochondrial Tricarboxylic Acid (TCA) Cycle

The TCA cycle catalyzes the complete aerobic oxidation of glucose that occurs in the mitochondria. Mitochondria are the center of tumor metabolism reprogramming as they integrate all bioenergetics, biosynthetic, and redox signaling functions [95]. In contrast to the initial misconception that cancer cells adopt aerobic glycolysis as a result of impaired mitochondrial function, recent advances in the field indicate that a majority of tumor cells maintain the capacity to produce energy through mitochondrial oxidative phosphorylation (OXPHOS) [96]. Emerging evidences have indicated the requirement of mitochondrial metabolism in various tumor types, including pancreatic cancer [74,97,98,99,100]. The majority of glucose in tumors under in vitro culture conditions is converted into lactate, which is accordant with the pervasive belief that glucose utilization in tumor cells switches from oxidative metabolism to glycolysis. However, recent in vivo tracing experiments using isotope-labeled glucose in GEM models and cancer patients indicate that tumor tissues exhibit enhanced glucose contribution to the TCA cycle compared to normal tissue [101,102,103,104,105], underscoring the importance of characterizing metabolism reprogramming in physiological environments. Importantly, these in vivo flux studies in lung and pancreatic caners revealed the direct contribution to the TCA cycle from lactate, which dominates the contribution from glucose [101,103,106]. While a study in the PDAC GEM model indicates that glutamine contributes more to the TCA cycle than glucose does, the contribution of glucose to the TCA cycle is actually mostly through lactate, which allows more efficient utilization of glycolytic intermediates to support biosynthesis and tumor growth [103].

## 4. Nutrient Salvage and Glucose Metabolism

Metabolomics analysis of human PDAC indicated that multiple metabolites related to glucose and glutamine are depleted in tumors compared to adjacent normal tissues, supporting the limited nutrient availability in PDAC cells, which is likely due to the dense stroma and poor tumor perfusion [93]. Therefore, PDAC cells are engaged in multiple salvages pathways, including nutrient recycling through autophagy and nutrient scavenging from the extra-cellular space through macropinocytosis to meet nutrient requirements [7,8]. As a common effector for both autophagy and macropinocytosis, lysosome is concordantly activated constitutively in PDAC cells [107]. Diverse substrates, including sugars [108], glutamine [109], amino acid [10,110,111], nucleosides [112,113], and fatty acids [114,115], are generated from the nutrient salvage pathways to fuel tumor growth.

Autophagy is a lysosome-mediated self-digestion process that is critical for the maintenance of cell viability under stress conditions [116]. It is highly activated in PDAC cells and its requirement for tumor maintenance has been well-established [7,117,118]. As a tightly regulated process, autophagy can be induced by a variety of stress signals, including nutrient deprivation and hypoxia [119], both of which are hallmarks of human PDAC. Glucose deprivation, which is commonly observed in PDAC [93,120], has been shown to activate AMPK, which in turn activates autophagy through the phosphorylation of Beclin 1, a key autophagy component [121,122]. However, it should be noted that basal autophagy in human PDAC cells is maintained at highly elevated level even under nutrient-rich conditions [7], implicating that autophagy may be constitutively activated in PDAC cells. It remains to be further evaluated whether tumor autophagy is further elevated by nutrient stress signals under in vivo conditions. On the other hand, recent studies indicate that autophagy activation leads to the induction of glycolysis [123,124]. However, the mechanism underlying autophagy-mediated glycolysis activation is still poorly understood.

Besides autophagic recycling, PDAC cells also scavenge nutrients by taking up the extracellular macromolecules through macropinocytosis. Oncogenic KRAS signaling is a potent inducer for macropinocytosis in PDAC cells [8,9]. In addition, a recent study indicates that macropinocytosis can also be activated by glutamine deprivation in PDAC [125]. It remains to be investigated whether additional stress signals, such as glucose deprivation or hypoxia, also induce macropinocytosis in a similar fashion. It is also not clear how macropinocytosis may affect the glucose metabolism programs in tumor cells. Recent studies largely focused on the salvage of extracellular protein or lipid as nutrient sources [8,93,126,127,128]. Given the abundance of complex carbohydrates in the extracellular space, additional efforts are warranted to investigate whether polysaccharides salvage through macropinocytosis or other uptake mechanisms may impact tumor cell glucose metabolism and oncogenic growth.

## 5. Glucose Metabolism in Intra-Tumoral Crosstalk

Human PDAC is a heterogeneous disease and recent large-scale transcriptomic analyses have classified PDAC into several molecular subtypes that show distinct clinical characteristics [129,130,131,132]. Besides the ADEX/exocrine and immunogenic subtypes, which are likely defined by signatures derived from non-neoplastic cells [130,132], the molecular signatures of cancer cells largely fall into two categories: The squamous/quasimesenchymal/basal-like subtype enriched with mesenchymal signatures and the progenitor/classical subtype enriched with epithelial signatures. Interestingly, PDAC cell lines or primary PDAC tumors of the mesenchymal subtype are associated with a more glycolytic phenotype and elevated expression level of MCT4 compared to those of the epithelial subtype [6,23], implicating distinctive metabolic dependencies among different subtypes of tumor cells. While the molecular subtypes reflect the inter-tumoral heterogeneity, recent single cell analyses of human PDAC have also identified sub-clusters of tumor cells exhibiting similar mesenchymal and epithelial signatures [133,134], although it remains to be investigated whether these sub-clusters of tumor cells are also associated with different metabolic characteristics within the bulk tumor. It is possible that the formation of such intra-tumoral heterogeneity of tumor cells is largely dictated by the local microenvironment, such as nutrient and oxygen availability. As previously described, PDAC is characterized with large areas of hypoxic regions, likely due to poor perfusion caused by the dense stroma [44]. A recent study showed that tumor cells in hypoxic regions tend to undergo epithelial–mesenchymal transition (EMT) and exhibit elevated glycolysis compared to tumor cells in the normoxic areas [84]. Importantly, a symbiotic relationship has been identified between hypoxic tumor cells and normoxic ones. MCT4 is highly induced in hypoxic PDAC cells to mediate the excretion of lactate derived from heightened glycolysis, which is taken up through another lactate transporter, MCT1, which is exclusively expressed in the normoxic tumor cells to fuel tumor growth [84]. Similar metabolic crosstalk between MCT4 high and MCT1 high tumor cells has also been described in other cancer types [135,136,137,138], implying that such metabolic symbiosis is a common theme in the tumor ecosystem. In addition to the metabolic heterogeneity amongst tumor cells of different molecular subtypes, PDAC tumor-initiating cells (TICs) also exhibit distinctive metabolic phenotypes. It has recently been shown that, compared to proliferating tumor cells, PDAC TICs are equipped with limited metabolic plasticity and exhibit decreased glycolysis activity accompanied with enhanced mitochondrial OXPHOS [99,100]. As a result, PDAC TICs feature unique dependence on mitochondrial respiration that can be therapeutically targeted.

The presence of exuberant stroma is one of the most defining features of human PDAC. Extensive metabolism crosstalk occurs between tumor cells and their microenvironment, including the exchange of nutrients between PDAC cells and associated fibroblasts (CAFs) [139,140]. It has been shown that CAFs have higher metabolic flexibility than normal fibroblasts and PDAC cells also increase the autophagy in CAFs [10,141]. The non-essential amino acids (mostly alanine) released from CAFs in turn fuel the TCA cycle of tumor cells to support anabolic needs, such as lipid biosynthesis [10]. As such, CAFs promote the consumption of glucose in PDAC cells and enhance the shunt of glucose into anabolic pathways, such as serine biosynthesis [10,142]. In addition, under conditions of glucose deprivation, pancreatic cancer cells uptake collagens via micropinocytosis, which provides proline as a nutrient source to fuel the TCA cycle and macromolecular biosynthesis [143]. Metabolic interactions in the microenvironment also function to suppress anti-tumor immunity to allow malignant progression. Specifically, lactate excreted by PDAC cells has been shown to inhibit natural killer (NK) cells’ function and reprogram macrophages into immunosuppressive cells [144,145]. As evidence accumulates about the metabolism interactions between the stromal and epithelial cell compartments in PDAC, it is becoming evident that co-targeting both the cancer cells and their microenvironment may contribute to improve clinical outcomes for patients.

## 6. Pancreatic Cancer and Diabetes

The initiation and progression of PDAC is also affected by systemic metabolism alterations, such as diabetes and obesity. There is a 1.5- to 2.0-fold increase in the risk of developing pancreatic cancer in type 2 diabetes mellitus patients [146,147]. The hyperglycemia, which is caused by insulin resistance and inability to suppress inappropriate hepatic glucose release, has been shown to enhance proliferation [148,149], promotes epithelial–mesenchymal transition and cancer stem cells’ properties [150], and metastatic potential in pancreatic cancer [151,152]. Recently, it has been reported that high glucose concentrations in the cell culture medium induce the *KRAS^G12D^* mutation in vitro. The induction of de novo oncogenic *KRAS* mutations is related to the lower PFK activity in pancreatic ductal cells and elevated O-GlcNAcylation under high-glucose conditions [153]. The compromised ribonucleotide reductase (RNR) activity by O-GlcNAcylation leads to a deficiency in deoxynucleotide (dNTP) pools, genomic DNA alterations, and consequently *KRAS* mutations [153]. It remains to be further validated whether the elevated blood glucose level in diabetes patients could indeed induce *KRAS* mutations in vivo. Additionally, obesity, which is often associated with diabetes, is also an independent risk factor of increased pancreatic cancer. In agreement with the epidemiologic data, a high-fat diet promotes the development of pancreatic intraepithelial neoplasias (PanINs) and PDAC by the induction of inflammation and fibrosis than the control diet [154,155]. On the other hand, it has been recognized that pancreatic cancer also plays a causal role in the development of a form of new onset diabetes, namely type 3c (pancreatogenic) diabetes [156]. In contrast to classic type 3 diabetes caused by exocrine pancreatic deficiency, such as chronic pancreatitis, which is characterized with low insulin and increased peripheral insulin sensitivity, PDAC-induced type 3c diabetes often exhibits high insulin levels and peripheral insulin resistance, similar to type 2 diabetes [157]. It has been shown in a case-control study of patients with PDAC that hyperglycemic may precede PDAC diagnosis for a mean period of 36 to 30 months, indicating such a new onset of diabetes may serve as an early detection marker for pancreatic cancer [158,159]. Besides diagnosis, targeting hyperglycemic may also benefit the PDAC prevention and treatment. Metformin, a well-known antidiabetic drug that inhibits the mitochondria electron transportation chain (ETC) [160,161], has been shown to be associated with increased survival among diabetic PDAC patients [162,163,164]. However, the beneficial effect of metformin in PDAC patients was called into question by a recent study showing a lack of correlation between metformin and PDAC patient survival [165], indicating additional in-depth studies are needed to understand the molecular mechanisms for the effect of metformin on tumor cells and systematic metabolism.

## 7. Conclusions and Future Directions

Good progress has been made in understanding reprogrammed metabolism in PDAC. The accumulated knowledge points to various unique and targetable vulnerabilities, such as in the rewired glycolytic flux and its branched pathways like PPP, HBP, serine biosynthesis, and TCA. Given the high plasticity of the metabolic pathways and complexity of metabolic crosstalk amongst tumor components, it is unlikely that we would achieve a meaningful response by targeting any single pathway. It is urgent to characterize the compensation among the metabolic pathways and also the metabolic dependence among various tumor components, which may help to identify new metabolism drug targets. Recent concerted efforts on the genomic and epigenomic profiling of tumors from both human and GEM models, in particular the recent advances in the field of single cell sequencing, have revealed the complexity of PDAC with unprecedented resolution. Such knowledge will enhance our understanding of metabolic crosstalk in the TME and may help identify effective strategies targeting metabolism in PDAC.

## Figures and Tables

**Figure 1 cancers-11-01460-f001:**
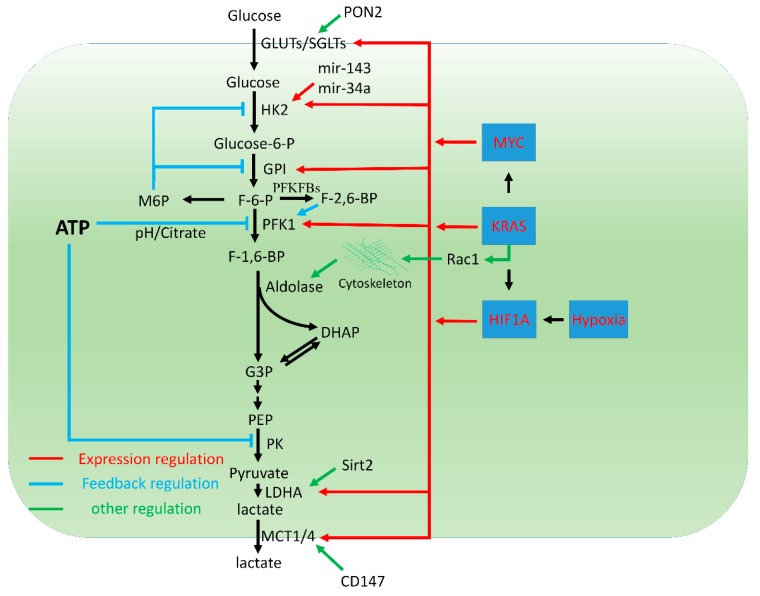
Regulation of glycolysis in PDAC.

**Figure 2 cancers-11-01460-f002:**
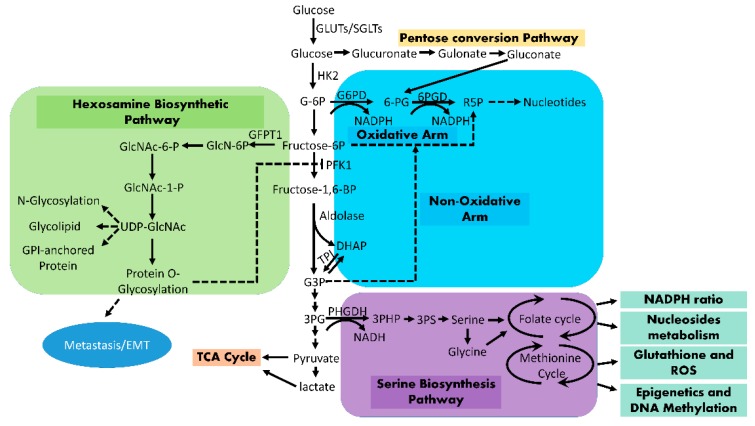
Enhanced glucose flux into anabolic pathways in PDAC.

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
