# Peer review of "Glucose Metabolism in Pancreatic Cancer"

_cancers, 2019, doi:10.3390/cancers11101460_

Round 1

Reviewer 1 Report

This paper by Yan et al. presented an overview of glucose metabolism in pancreatic cancer. The content includes clear descriptions of glycolysis, enhanced glucose flux, nutrient salvage and glucose metabolism, and glucose metabolism in intra-tumoral crosstalk. Overall this review is well written and gave a good overview on recent progresses made in understanding glucose metabolism-related deregulations in PDAC. The review is worthy of publication after including some key information which will further improve the quality and integrity of the review.

Comments:

The effect of aberrant glucose metabolism induced by diabetes or diet in PDAC could be included in this review. The authors may draw a model to describe how deregulation of glucose metabolism affects PDAC growth or metastasis. The authors may summary the current treatment against glucose metabolism in PDAC. The recent report (Hu, et al. Cell metabolism 2019) clearly showed that a mechanistic link between a perturbed sugar metabolism and genomic instability that induces de novo oncogenic KRAS mutations and triggers cell transformation preferentially in pancreatic cells. This paper should be included in the review.

Author Response

The effect of aberrant glucose metabolism induced by diabetes or diet in PDAC could be included in this review.

We thank the reviewer's constructive comments. In the revised manuscript, we included additional paragraph discussing the relationship between diabetes and PDAC (line 400-430).

The authors may draw a model to describe how deregulation of glucose metabolism affects PDAC growth or metastasis. 

In contrast to the vast knowledge gained on the metabolism reprogramming in primary PDAC, little is known regarding the metabolism alterations in PDAC metastasis, though we included the recent identification of unique dependency on PPP in metastatic tumor cells (line 240-247). Therefore, would agree that it is difficult to draw an informative diagram on the role of glucose metabolism in PDAC progression and metastasis.

The authors may summary the current treatment against glucose metabolism in PDAC. 

We agree with the reviewer that this is an important point. There are several agents targeting glucose metabolism such as LDH and MCT inhibitors are under preclinical development and early phase clinical trials. We directed readers to a recent review by Luengo et al which nicely summarized the progress in the targeting metabolism pathways for cancer therapy.

The recent report (Hu, et al. Cell metabolism 2019) clearly showed that a mechanistic link between a perturbed sugar metabolism and genomic instability that induces de novo oncogenic KRAS mutations and triggers cell transformation preferentially in pancreatic cells. This paper should be included in the review.

We thank reviewer pointing this out. We now included this important study in the session about diabetes and PDAC.

Reviewer 2 Report

I reviewed the review entitled "Glucose Metabolism in Pancreatic Cancer" by Yan et al.

The manuscript is well written and well organized.  I only have the following comments:

1) Line 34 add "the"    .... as "the" Warburg effect.

2) Line 55 add "of the"  ... a broad description "of the" reprogramming

3) Line 69 add the other nomenclature for Glut1 and Glut5 ... SLC2A1 and SLC2A5, SGLT2, etc

4) The only thing that seems missing is a short summary of the analytical methods used to analyze glucose metabolism

Author Response

We thank the reviewer's positive comments.

1) Line 34 add "the" .... as "the" Warburg effect.

We made the correction as suggeted.

2) Line 55 add "of the" ... a broad description "of the" reprogramming

We made the correction as suggested (line 67 in the revised version).

3) Line 69 add the other nomenclature for Glut1 and Glut5 ... SLC2A1 and SLC2A5, SGLT2, etc

We included the official gene nomenclature for Glut-1 and SGLT2 in the revised manuscript (line 81 and 96 respectively).

4) The only thing that seems missing is a short summary of the analytical methods used to analyze glucose metabolism

We agree with reviewer that this is an important point to be included. We included a sentence briefly described the commonly used analytic tools for metabolism studies that contributed to our current knowledge on metabolism reprogramming (line 34-38). However, we didn't get into any details due to our lack of in-depth knowledge on the technical details of those analyses.

Reviewer 3 Report

Title: Glucose metabolism in Pancreatic Cancer.

Authors: Liang Yan et.al

This review discusses the basis and consequences of alterations in glucose metabolism in pancreatic cancer. The article covers relevant topics related to the unique biology of pancreatic cancer (Panc Ca) i.e. Kras mutations and micro-environment driven hypoxic and hypo-vasculature-induced changes in metabolism. The reader will benefit from greater mechanistic insight into how elevated glucose metabolism promotes for example drug resistance and alternative metabolic dependencies. Several statements are cursory and lack in depth review the of context specific findings supporting the observation. Please see comments below:

Major comments:

What are the subtypes of Panc Ca and how do their metabolic states differ (alluded to in lines 48-50) Facilitative transport of glucose is not practically bi-directional as it gets phosphorylated by hexokinase (which is particularly relevant for a cancer cell). How does PON2 regulate GLUT1 expression? This is a unique biology in Panc Ca that you can delve in to more mechanistically. Glycolysis comprises 10 steps not 12 and to sustain glycolysis. Apart from product accumulation one needs to maintain NAD production to sustain glycolysis. (Line 92 and 98) How does mutant Kras elevate GLUT1 and furthermore promote elevated anabolic non-oxidative metabolism of glucose? You however mention the oxidative pathway (Line 190). Examination of the Ying, H 2012 paper is fundamental to this switch and proposes increased dependency on the non-oxidative arm. In this context of Kras mutations you can also discuss why glutamine metabolism, and other nutrient scavenging/uptake pathways are elevated. What is the mechanistic basis for elevation of the lactate transporters and how does circulating lactate import sustain the TCA? What are the roles of MUC and HIF1 in promoting glucose metabolism and resistance in Panc Ca? Again mechanistic insight into the important findings that support suppression of nucleotide synthesis and resistance to gemcitabine are very relevant to Panc Ca. FBW7 regulates c-Myc and TXNIP to suppress glucose metabolism – what is its significance in Panc Ca and glucose anabolism? Connections between glucose metabolism and metastasis – what are the supporting pathways? What are the translational implications: in diagnostics and in diabetes.

Minor comments:

Line 12: the inability to target pancreatic cancer may not entirely be due to a poor understanding but more due to resistance pathways including metabolic plasticity. The first paragraph of the introduction is somewhat a regurgitation of the abstract. Line 150 – glycolysis genes

Author Response

We thank the reviewer's constructive comments.

What are the subtypes of Panc Ca and how do their metabolic states differ (alluded to in lines 48-50)

In the revised manuscript, we included a brief description of the metabolism subtypes of PDAC in line 55-58.

Facilitative transport of glucose is not practically bi-directional as it gets phosphorylated by hexokinase (which is particularly relevant for a cancer cell). 

We thank the reviewer for pointing this out. We deleted the 'bi-directional' in line 79.

How does PON2 regulate GLUT1 expression? This is a unique biology in Panc Ca that you can delve in to more mechanistically. 

We now included additional details on PON2-mediated regulation of GLUT1 in line 92-94.

Glycolysis comprises 10 steps not 12 and to sustain glycolysis. 

We used 12 steps because we included glucose uptake and lactate excretion. We agree with the reviewer's comment and changed it to 10 steps to precisely reflect the actual steps of glycolysis.

Apart from product accumulation one needs to maintain NAD production to sustain glycolysis. (Line 92 and 98) 

We thank the reviewer for pointing this out. We now included the discussion on the role of NAD production on glycolysis in line 130-133.

How does mutant Kras elevate GLUT1 and furthermore promote elevated anabolic non-oxidative metabolism of glucose? You however mention the oxidative pathway (Line 190). Examination of the Ying, H 2012 paper is fundamental to this switch and proposes increased dependency on the non-oxidative arm. 

KRAS regulation the expression of glycolysis genes including GLUT1 at transcriptional level through MYC-dependent mechanism (line 166-179). The activation of non-oxidative PPP by KRAS is also likely due to the transcription of RPIA and RPE (line 254-256). The mention of the oxidative PPP was referred to the earlier studies on KRAS-induced cellular transformation experiments. We described the findings from our 2012 paper on KRAS-driven non-oxidative PPP in more detail in line 252-258.

In this context of Kras mutations you can also discuss why glutamine metabolism, and other nutrient scavenging/uptake pathways are elevated.

We appreciate the reviewer's suggestion. Since our review is more focused on glucose metabolism, we briefly mention the KRAS-induced glutamine metabolism in the introduction par (line 54-55). We discussed the relationship between glucose metabolism and nutrient salvage pathways in more detail in 325-359.

What is the mechanistic basis for elevation of the lactate transporters and how does circulating lactate import sustain the TCA? 

KRAS has been shown to induce the expression of MCT4. In addition, CD147 has been shown promote the membrane localization of MCT1 and MCT4. We included these in line 128-136. While recent studies indicates lactate directly support TCA, the molecular mechanisms for the transportation of lactate into mitochondria and its utilization in TCA remain to be elucidated.

What are the roles of MUC and HIF1 in promoting glucose metabolism and resistance in Panc Ca?

We provided additional mechanistic details for the role of MUC on HIF1 stabilization in line 180-181.

FBW7 regulates c-Myc and TXNIP to suppress glucose metabolism – what is its significance in Panc Ca and glucose anabolism?

We included the discussion on the role of FBW7 on glucose metabolism in line 202-205.

Connections between glucose metabolism and metastasis – what are the supporting pathways? 

Unfortuantely, very little is known on the role of glucose metabolism and metastasis. We included the discussion on the dependency on PGD in metastasis in more detail (line 244-251).

What are the translational implications: in diagnostics and in diabetes. 

We now added an additional paragraph discussion the relationship between PDAC and diabetes, which also included the diagnostic and therapeutic implication (line 408-437).

Line 12: the inability to target pancreatic cancer may not entirely be due to a poor understanding but more due to resistance pathways including metabolic plasticity. 

We thank the reviewer's insightful comment. we included 'metabolic plasticity' in the revised version (line 12).

Line 150 – glycolysis genes 

We corrected the mistake (line 185).

Reviewer 4 Report

This is a nicely written review summarizing glucose metabolism in pancreatic cancer. The topic is timely and the literature citing is thorough and balanced. I enjoyed reading it.

Author Response

We thank the reviewer's positive comments.

Round 2

Reviewer 3 Report

All comments have been addressed. Just few minor points:

Line 12 – it is somewhat odd to insert metabolic plasticity there with no other context. The original intent was to expound on what are the main PDAC-specific features related to resistance. In lieu of this it is best to leave it that sentence in the original form.

Line 151: “The glycolytic activity in tumor cells is concordant with the transcriptional regulation of glycolytic genes” this is not entirely true – gene expression does not necessarily correlate with activity. You could modify to state that it can be "in part regulated by expression of rate-limiting glycolytic genes"?

Author Response

Line 12 – it is somewhat odd to insert metabolic plasticity there with no other context. The original intent was to expound on what are the main PDAC-specific features related to resistance. In lieu of this it is best to leave it that sentence in the original form.

We have made the change as reviewer suggested.

Line 171: “The glycolytic activity in tumor cells is concordant with the transcriptional regulation of glycolytic genes” this is not entirely true – gene expression does not necessarily correlate with activity. You could modify to state that it can be "in part regulated by expression of rate-limiting glycolytic genes"?

We thank reviewer for the suggestion. We have made the correction as reviewer suggested.